# High-Speed Rail and Urban Growth Disparity: Evidence from China

**Haoran Zhang** [1] , **Ying Chai** [1,*], **Xuyu Yang** [2] **and Wenli Zhao** [1]

1   School of Economics, Guangdong University of Finance and Economics, Guangzhou 510320, China; zhr20@126.com (H.Z.); wenli_zzb@163.com (W.Z.)
2   School of Sociology and Population Studies, Renmin University of China, Beijing 100872, China; beginyang@163.com
*   Correspondence: chaiying19@163.com

**Abstract:** We investigate the effects of high-speed rail (HSR) operation on urban growth disparity in China. Using urban panel data from 2005 to 2019 and difference-in-differences estimation, we find that the operation of HSR has exerted a strong and robust positive effect on urban growth and total factor productivity (TFP) in core cities, while this effect is much weaker in non-core cities, especially in non-core cities close to provincial capitals. Meanwhile, high-speed rail has triggered relatively slower urban growth in the central cities compared with the suburbs in regional primate prefectures. The results suggest that the extension of HSR promotes centralization across cities and local decentralization within regional primate cities.

**Keywords:** high-speed rail; urban growth; TFP; centralization across cities; decentralization within cities

## 1. Introduction

Since the operation of the Beijing–Tianjin intercity high-speed rail in 2008, China's HSR system has entered a stage of rapid development. By 2021, the operational mileage of HSR in China exceeded 40,000 km, more than the sum of the operating mileage of all other countries combined, making it the world's largest and longest dedicated passenger railway project. The HSR has the advantages of high speed, low pollution, and large capacity, which may not only improve the flow of people, capital, and technology but also lead to the redistribution of resources and economic activity. Several regional and urban plans and policies emphasize the development of hinterland areas and suburbs through improving the transportation infrastructure. Therefore, understanding the spatial distributional effect of enhanced passenger transport links is particularly important for policy makers. Although some studies have considered the heterogeneous impacts of HSR [1–3], few papers discuss the effect of HSR on the growth disparity and location of economic activity within and across cities.

In theory, new economic geography (NEG) models predict that economic activity will proceed from decentralization to agglomeration with the decline in the transportation costs of goods between large areas [4–6], whereas the classic monocentric model in urban economics [7,8] suggests that the reduction of commuting time within cities promotes the decentralization of a city. HSR mainly reduces the transportation costs of people and ideas across cities, which is not completely consistent with the context of NEG models or urban economic models. Meanwhile, the question of how to link the theoretical traffic cost parameters with reality still requires rigorous empirical analysis. Therefore, it is of great theoretical and policy value to examine the impacts of HSR operation on China's growth disparity and economic geography.

This paper takes the operation of HSR as a quasi-experiment and estimates its impacts on urban growth using a difference-in-differences (DID) framework. The results show that the expansion of HSR promotes the centralization of urban systems and decentralization

within core cities in China. Our results are robust to several specification checks, such as alternative measures of HSR and urban growth.

This paper is related to a few recent empirical works that consider the heterogeneous effects of HSR [1–3]. Specifically, Diao [1] finds that the effect of HSR on fixed-asset investment depends on city size and that second-tier cities benefit more. Jin et al. [2] find heterogeneous impacts of HSR depending on city size and development level. Jiao et al. [3] find that the impacts of accessibility and connectivity vary with geographic region and city size. The differences between our paper and previous work include the following: First, we investigate the effects of HSR on the growth disparity and economic geography within and across cities. Differences in geographical scales matter when it comes to the spatial distributional effect of transportation infrastructure. The competition for land and housing is much fiercer within cities. High-speed rail may lead to opposite economic geography consequences within and across cities. Second, we also examine the heterogenous effects of high-speed rail by non-core cities close to municipalities and provincial capitals, rather than only distinguishing them by geographic regions or by endogenous city size, which has been determined in part by high-speed rail. Third, Diao [1], Jin et al. [2] and Jiao et al. [3] focus on level effects. In this paper, we focus on the growth effect, which is more suitable to reflect changes in share and economic geography.

The remainder of this paper is organized as follows. Section 2 provides the literature review. Section 3 presents the conceptual framework. Section 4 details the model, data and variables. Section 5 discusses the results. Section 6 concludes.

## 2. Literature Review

In a groundbreaking study [4] and subsequent related research, New Economic Geography models suggest that the decline in transportation costs may increase the population of the core area at the expense of population loss in the periphery area. The idea of these NEG models is that the local market effect amplifies with population inflow. Manufacturers in periphery areas enjoy a certain degree of protection when transportation costs are high. As transportation costs decline, residents in periphery areas can import certain goods from the core that were supplied previously by local producers, resulting in a shift of employment from the periphery areas to the core areas. Baum-Snow et al. [6] argue that the context of such models applies well to China, where the cost of inter-regional population mobility is high.

The literature in the field of international trade and NEG focuses on the transportation costs of goods between large areas, whereas urban economics emphasizes the transportation costs of people within cities. In the classic monocentric model [7,8] in which most workers commute to the city center, the shortening of travel time within cities reduce the relative value of locations near the city center [9], thereby increasing the space demand of the suburbs, in turn promoting the suburbanization and decentralization of the city. This conclusion is contrary to the prediction of the New Economic Geography models. On one side, the high-speed rail is a passenger-dedicated railway, which mainly reduces the residents' transportation costs and is consistent with the context of urban economics; on the other side, high-speed rail mainly reduces transportation costs between cities, which is similar to the context of the NEG model. In this sense, the existing theoretical models do not reveal whether the HSR promotes the centralization or decentralization of economic activity.

Since Baum-Snow [10], academia has begun to estimate the effects of highway and subway on the spatial structures within cities. The results indicate that transportation infrastructure improvements can lead to decentralization within cities [9–11].

Another strand of literature involves the impacts of highways and ordinary railways on the economic geography and spatial structure of urban systems; however, no consistent conclusions have been reached. As far as China is concerned, Faber [12] finds that highways may be detrimental to urban growth in counties more than 50 km distant from core cities. Qin [13] finds that the affected counties experienced reductions in GDP following ordinary railway upgrades. Baum-Snow et al. [6] further find that highways promote the

economic and population growth of regional primate cities at the expense of hinterland cities. However, Banerjee et al. [14] show that better access to roads and railways is beneficial to ordinary Chinese counties.

Although the impact of HSR has been studied extensively, previous works have largely focused on comparing the spatial configurations in HSR networks, airline networks, or inter-city coach networks [15,16] or have discussed the effects of HSR on housing and land prices [17,18]; access, connectivity, and centrality [19,20]; aviation [21]; urban growth [22]; land-use efficiency [23,24]; regional equity [25]; city size [26]; knowledge spillover [27]; innovation performance [28]; CO2 emissions [29]; entrepreneurship [30]; ecological environment pressure [31]; and regional economic sustainability [32]. Existing studies have ignored the effects of HSR on the growth disparity within and across cities.

As passenger-dedicated lines between cities, HSR represents one of the most important transportation technology breakthroughs in recent decades. Its impacts on growth disparity and economic geography should be different from those of passenger and cargo mixed roads, highways, and ordinary railways. The focus on HSR excludes the effect of cargo transportation, and it is more conducive to identifying the mechanisms. Therefore, we estimate the impacts of HSR on the GDP growth within and across cities using the DID strategy.

## 3. Conceptual Framework

### 3.1. HSR Effects across Cities

High-speed rail can quickly transport consumers, workers and soft information; help to reduce travel times between headquarters and affiliates [33]; and lower face-to-face communication and sales costs with external stakeholders as well as equipment installation, maintenance and other service costs.

Productivity in core cities is high relative to non-core cities due to greater agglomeration [34]. The expansion of HSR enables firms in core cities to connect with new suppliers and customers and facilitates the inflows of labor and capital. Therefore, the effect of HSR on urban growth may be larger in core cities.

For non-core cities with high-speed rail connections, the role of transport infrastructure improvements may be mixed. On the one hand, the extension of HSR reduces the costs of doing business and facilitates firms in non-core cities in reaching new markets. On the other hand, the extension of HSR may expose firms in non-core cities to fiercer competition [35], transfer support activities and high skilled managers to headquarters in core cities [33] and make it easier for workers from small cities to gain access to external large markets, thus enhancing the relative advantage of the core cities.

### 3.2. HSR Effects within Core Cities

Theoretically, high-speed rail may trigger local decentralization within core cities. The competition for land and housing should be particularly severe within core cities. This feature is not considered by the NEG models [4,5]. New Economic Geography models typically focus on large areas and neglect the effects of land markets. By contrast, land competition is critical in urban economics, which leads to the dispersion of economic activity with the reduction of transportation costs [36]. Additionally, durable housing capital and expensive demolition costs may hinder growth in the central city. High-speed rail stations are also more likely to be built in the suburbs to save costs [20], which may play a role in the process of decentralization within core cities.

## 4. Estimation Strategy

### 4.1. Model Specification

The main goal of our paper is to discuss the effect of HSR expansion on the growth disparity and spatial distribution patterns of economic activity. Considering that GDP growth can comprehensively reflect urban population growth, output growth, and productivity growth, we take the real GDP growth rate as our primary outcome. We also consider urban

growth in real GDP per capita, growth in urban employment, fixed asset investment and TFP for a robustness check and further analysis.

If high-speed rail operations make core cities' or prefectures' central areas grow faster than non-core cities or prefecture remainders, the economic or employment share in core cities' or prefectures' central areas will increase. This means that core cities' or prefectures' central areas become winners in the process of HSR extension, and HSR operations promote the process of centralization of economic activity; conversely, it means that HSR operations promote the decentralization of economic activity. Specifically, we construct the following equation:

$$y_{it} = \alpha + \delta_i + \lambda_t + \beta_1 HSRail_{it} + x_{it}\gamma + \varepsilon_{it} \tag{1}$$

where $HSRail_{it}$ is our regressor of interest, which is a high-speed rail operation dummy variable in most regressions. Given that most HSR lines enter operation in the second half of the year and that the year in which they open is usually a trial operation stage, we take the years after the opening of an HSR system as one and introduce the dummy variable (Open) for the year when the HSR opened to control for the influence of the opening year. Inspired by Navarro and Turner [11], we also count the annual number of urban operational HSR lines as a robustness check. As the number of operational high-speed rail lines is time-point data, we average the numbers at the beginning and end of the year in each city: $x_{it}$ is the set of controls, $\delta_i$ is the city fixed effects, $\lambda_t$ is the year fixed effects. To investigate the heterogenous effects of HSR, we introduce the interaction terms of high-speed rail with core cities (Core) and non-core cities (Noncore):

$$y_{it} = \alpha + \delta_i + \lambda_t + \beta_1 CoreHSRcities_{it} + \beta_2 NoncoreHSRcities_{it} + x_{it}\gamma + \varepsilon_{it} \tag{2}$$

The scope of the core cities (regional primate cities) includes the municipalities and the provincial capitals (PCs) in turn in different models [16].

### 4.2. Data and Variables

To examine the HSR effects within and across cities, we use 285 prefecture-level cities with urban and rural areas (total city) and prefectures with urban areas (districts under city). Figure 1 shows the study area, which covers almost 90% of China's population [6,37]. The time span of our data is from 2005 to 2019. Before the opening of the Beijing–Tianjin high-speed railway in August 2008, the Qinhuangdao–Shenyang passenger dedicated line opened bullet trains in 2005 and was classified as a high-speed railway by the China Railway Corporation. The China City Statistical Yearbook does not provide GDP level, GDP growth rate or a number of controls within urban and rural areas for 2017, so we exclude the sample of the total city in 2017. Due to frequent changes in the scope of urban areas, we drop the districts under cities with a change in land area of more than 1% each year in most regressions. The main data for this paper come from the China City Statistical Yearbook. GDP is adjusted to the real price in 2005 using the GDP deflator index. The information about the high-speed railway lines and stations comes from the Chinese Research Data Services (CNRDS) platform (https://www.cnrds.com/Home/Login, accessed on 25 December 2021). The main criterion for judging a HSR station is whether the station is located along a passenger-dedicated railway line [38] with a design speed of not less than 250 km/h [39]. As a robustness check, this paper also uses a more relaxed definition that includes ordinary railways with C, D and G pre-fix bullet trains [16,19] and a more rigorous definition that includes train services with peak speeds of over 300 km/h [1].

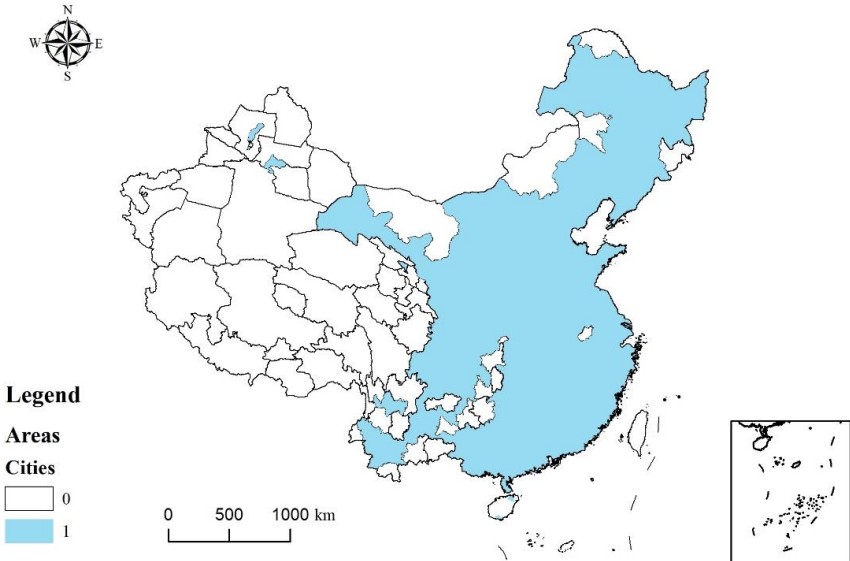

**Figure 1.** Study area. (Source: Author; the original data are from the National Geomatics Center of China).

The city-level control variables include elevation (average elevation in meters) and slope (average slope in degrees), both of which are extracted from the DEM of the SRTM with 90 m resolution; latitude, which is extracted from the National Geomatics Center of China; investment rate (Inv): given that high-speed rail may affect economic growth through physical investment, we control the preexisting percentage of fixed asset investment in GDP in 2004; human capital (Edu), measured by the urban average years of education in 2000, sourced from the population census for 2000. All of the above preexisting control variables are included in the DID estimations, interacted with year dummies to flexibly control for the time effects of the baseline characteristics on urban growth.

Other city-level control variables include: GDP with a year lag, or GDP per capita with a year lag, or urban employment with a year lag in different regressions, depending on the outcomes, which are used to control initial conditions; subway, which we measure by the natural logarithm of subway mileages with a year lag [11,37]. The subway information is obtained from Wikipedia and the urban rail network (http://www.urbanrail.net/, accessed on 21 December 2021); and industrial structure, the percentages of the value-added of secondary industry or tertiary industry to GDP. Descriptive statistics of our main variables are shown in Table 1.

**Table 1.** Descriptive statistics.

| Variable | Mean | Std. Dev. | Min | Max |
|---|---|---|---|---|
| GDP growth rate | 11.10 | 4.36 | −19.38 | 37.00 |
| HSRail | 0.25 | 0.43 | 0 | 1 |
| HSRail lines | 0.37 | 0.69 | 0 | 4.5 |
| Core city | 0.11 | 0.31 | 0 | 1 |
| $\log (GDP_{t-1})$ | 15.90 | 1.03 | 12.92 | 19.40 |
| Open | 0.05 | 0.21 | 0 | 1 |
| Subway | 0.24 | 1.00 | 0 | 6.51 |
| Physical investment rate | 42.22 | 15.38 | 13.08 | 108.26 |
| Education | 7.66 | 0.76 | 5.08 | 9.99 |
| Latitude | 32.92 | 6.68 | 18.23 | 50.24 |
| Slope | 7.50 | 5.36 | 0.39 | 24.71 |
| Elevation | 517.61 | 598.58 | 1.33 | 3103.05 |
| Share of industry in GDP | 47.92 | 11.16 | 9.00 | 90.97 |
| Share of services in GDP | 38.58 | 9.65 | 8.58 | 85.34 |

## 5. Results

### 5.1. Baseline Results

To avoid the estimation accuracy being overestimated due to heteroscedasticity and serial correlation, the standard errors are clustered at the prefecture-level city. The baseline results from Equation (1) (column 1) and Equation (2) (columns 2 to 8) are reported in Table 2. Column (1) shows that the HSR operation has increased cities' real GDP growth rate by 0.52 percentage points. The estimate is significant at the 5% level. The coefficient is large in magnitude, given that a city's average GDP growth rate is about 10 percentage points.

**Table 2.** Baseline regressions.

| | (1) | (2) | (3) | (4) | (5) | (6) | (7) | (8) |
|---|---|---|---|---|---|---|---|---|
| | GDP Growth Rate | GDP Growth Rate | GDP Growth Rate | GDP Growth Rate | Per Capita GDP Growth Rate | Employment Growth Rate | Log (fixed Asset Investment) | TFP |
| HSRail | 0.523 ** (0.251) | | | | | | | |
| Municipality × HSRail | | 2.391 ** (1.188) | | 2.665 ** (1.184) | | | | |
| Provincial capital × HSRail | | | | 1.689 *** (0.458) | | | | |
| Core city × HSRail | | | 1.823 *** (0.464) | | 2.163 *** (0.685) | 3.619 (2.666) | 0.064 (0.051) | 1.528 ** (0.670) |
| Non-Municipality × HSRail | | 0.497 ** (0.251) | | | | | | |
| Non-core city × HSRail | | | 0.332 (0.257) | 0.336 (0.257) | 0.089 (0.375) | 1.924 (1.241) | 0.015 (0.029) | −0.174 (0.364) |
| log (GDP$_{t-1}$) | −15.371 *** (1.484) | −15.328 *** (1.477) | −15.570 *** (1.453) | −15.529 *** (1.457) | | | 0.533 *** (0.164) | −9.305 *** (1.916) |
| log (per capita GDP$_{t-1}$) | | | | | −16.328 *** (1.578) | | | |
| log (urban employment$_{t-1}$) | | | | | | −41.939 *** (3.112) | | |
| Open | −0.026 (0.205) | −0.010 (0.204) | 0.005 (0.204) | 0.011 (0.204) | −0.080 (0.279) | 3.006 * (1.641) | −0.006 (0.019) | 0.311 (0.299) |
| Subway | 0.345 *** (0.120) | 0.330 *** (0.119) | 0.216 * (0.118) | 0.220 * (0.118) | −0.071 (0.153) | 1.919 *** (0.569) | −0.045 *** (0.016) | 0.375 *** (0.140) |
| Share of industry in GDP | 0.313 *** (0.041) | 0.311 *** (0.041) | 0.309 *** (0.041) | 0.308 *** (0.041) | 0.423 *** (0.056) | 0.621 *** (0.177) | 0.021 *** (0.005) | 0.257 *** (0.071) |
| Share of services in GDP | 0.136 *** (0.042) | 0.134 *** (0.042) | 0.133 *** (0.042) | 0.132 *** (0.042) | 0.272 *** (0.062) | 0.544 ** (0.225) | 0.004 (0.006) | 0.214 *** (0.074) |
| Control × Year dummy | Yes | Yes | Yes | Yes | Yes | Yes | Yes | Yes |
| Constant | 216.798 *** (21.353) | 215.824 *** (21.264) | 218.360 *** (20.932) | 217.713 *** (20.999) | 127.401 *** (14.176) | 55.794 ** (22.207) | 2.983 (2.497) | 134.401 *** (27.652) |
| Observations | 3987 | 3987 | 3987 | 3987 | 3987 | 3987 | 3420 | 3420 |
| Within-$R^2$ | 0.711 | 0.712 | 0.713 | 0.713 | 0.530 | 0.196 | 0.893 | 0.378 |

Notes: Standard errors in parentheses are clustered by prefecture-level city. *** $p < 0.01$, ** $p < 0.05$, * $p < 0.1$.

Columns (2) to (4) introduce the interaction terms between HSR and a certain city type, discuss the heterogenous impact of HSR at different levels of the urban hierarchy, and estimate which cities will benefit more from HSR connections. Column (2) suggests a large and significantly positive effect for municipalities. The regression coefficient is approximately five times that of the non-municipality. Column (3) introduces the core city dummy variables. The core cities include municipalities and provincial capitals. The results indicate that the high-speed rail increases the GDP growth rate of the core cities by 1.82 percentage points, and the estimate is statistically significant at the 1% level, whereas the regression coefficient of non-core city is only 0.33 percentage points. Column (4) decomposes the core cities into municipalities and provincial capitals, with regression coefficients of 2.67 and 1.69, respectively. The high-speed rail operation effect for other cities is relatively small and statistically insignificant. The results suggest that the effect of HSR operation in different cities is obviously weakened in the order of municipalities, provincial capitals, and other cities.

In column (5), we investigate the effect of HSR on the growth in real GDP per capita, which is the measure of "people-based" economic growth. The interaction coefficients are slightly larger for core cities and slightly smaller for non-core cities compared with the increase in total GDP growth rate, suggesting that HSR operations enlarge the economic disparity between core cities and non-core cities.

Column (6) then estimates the effect of HSR on the growth in urban employment. The coefficients are positive for both core HSR cities and non-core HSR cities, with a point estimate of larger magnitude for core HSR cities. Column (7) also shows that the positive effect of HSR on fixed asset investment is larger in core cities compared with non-core cities.

Column (8) uses TFP as the dependent variable. The estimate is positive and highly statistically significant in core HSR cities, whereas the impact of HSR in non-core cities is not significant, with a negative and small magnitude.

### 5.2. Robustness Checks
#### 5.2.1. Parallel Trends

To check the common trend assumption required by a DID analysis, we replace the regressor $HSRail_{it}$ in Equation (1) with an array of dummy variables to capture the year-by-year effects of HSR on GDP growth along the lines described by Beck et al. [40]. Given that the HSR from Beijing to Tianjin was opened in 2008, and we drop the sample of the total city in 2017 due to data limitations; we consider a window spanning from 4 years before the operation of HSR until 8 years after the operation of HSR. Specifically, we estimate the following equation:

$$y_{it} = \alpha + \delta_i + \lambda_t + \beta_1 H_{it}^{-4} + \beta_2 H_{it}^{-3} + \cdots + \beta_{12} H_{it}^{8} + x_{it}\gamma + \varepsilon_{it} \tag{3}$$

where $H^{-j}$ equals 1 for states in the $j$th year before the operation of HSR and $H^{+j}$ equals 1 for states in the $j$th year after the operation of HSR. The year of operation (year 0) is used as the reference year. At the end points, $H_{it}^{-4}$ equals 1 for years that are 4 or more years before the operation of high-speed rail, while $H_{it}^{8}$ equals 1 for years that are 8 or more years after the operation of high-speed rail. Figure 2 plots the estimates for the GDP growth rate along with the 95% confidence interval, adjusted for city-level clustering.

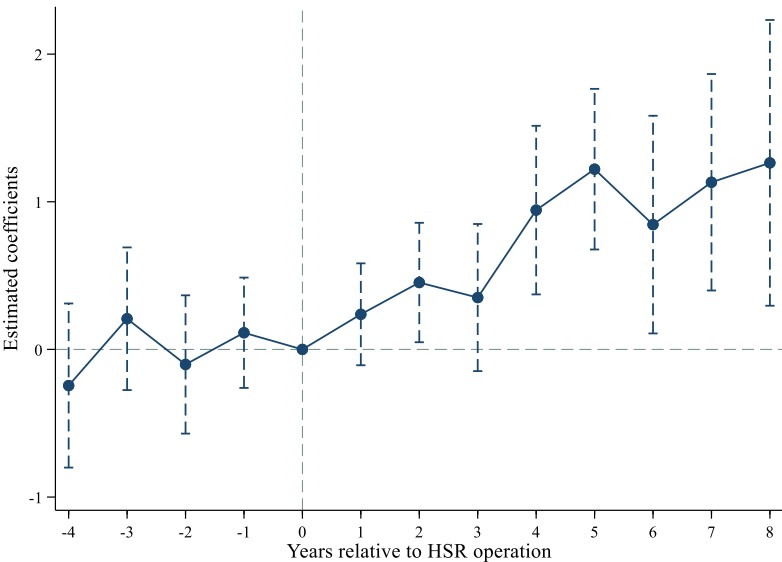

**Figure 2.** The dynamic effect of HSR on GDP growth. (Source: Author).

Figure 2 illustrates that the estimated coefficients are small in magnitude and insignificant for all years before the operation of HSR, suggesting a common trend between high-speed rail cities and non-HSR cities before the treatment. Figure 2 also allows us

to investigate the dynamic effects of HSR. We find that the estimates are positive and gradually increase in magnitude after the operation of HSR.

### 5.2.2. Parallel Trends Alternative Measurements of HSR

We use the annual count of HSR lines in each city to construct our alternative measure of high-speed rail, capturing more variation in the data. Table 3 reports the results. Column (1) indicates that one additional high-speed rail line is associated with a 0.94 percentage point increase in the urban GDP growth rate. Column (2) then estimates separate interaction coefficients between the number of HSR lines and dummies for whether the city is a core city. Once again, we find a sizable and significantly positive effect for core cities, while the interaction coefficient for non-core cities is insignificant and relatively small. Column (3) further considers the heterogeneous effects across municipalities and provincial capitals; the results show that the effect of HSR is larger in municipalities.

**Table 3.** Robustness checks.

| | (1) | (2) | (3) | (4) | (5) | (6) |
|---|---|---|---|---|---|---|
| | GDP Growth Rate | GDP Growth Rate | GDP Growth Rate | GDP Growth Rate | GDP Growth Rate | GDP Growth Rate |
| HSRail lines | 0.940 *** (0.298) | | | | | |
| Core city × HSRail lines | | 2.138 *** (0.453) | | | | |
| Municipality × HSRail lines | | | 2.667 *** (0.974) | | | |
| Provincial capital × HSRail lines | | | 1.998 *** (0.434) | | | |
| Non-core city × HSRail lines | | 0.605 * (0.319) | 0.603 * (0.319) | | | |
| Core city × HSRail | | | | 1.937 *** (0.459) | 1.900 *** (0.462) | 1.772 *** (0.456) |
| Non-core city × HSRail | | | | 0.215 (0.251) | 0.666 ** (0.300) | 0.520 ** (0.245) |
| log (GDP$_{t-1}$) | −15.457 *** (1.472) | −15.647 *** (1.446) | −15.613 *** (1.449) | −15.602 *** (1.450) | −15.513 *** (1.467) | −9.714 *** (1.046) |
| Open | | | | 0.018 (0.197) | 0.199 (0.205) | 0.112 (0.201) |
| Subway | 0.264 ** (0.120) | 0.089 (0.121) | 0.101 (0.119) | 0.193 (0.118) | 0.144 (0.116) | 0.279 ** (0.116) |
| Share of industry in GDP | 0.313 *** (0.041) | 0.307 *** (0.041) | 0.307 *** (0.041) | 0.307 *** (0.041) | 0.309 *** (0.041) | 0.266 *** (0.034) |
| Share of services in GDP | 0.134 *** (0.042) | 0.130 *** (0.042) | 0.129 *** (0.042) | 0.133 *** (0.042) | 0.135 *** (0.042) | 0.120 *** (0.035) |
| Control × Year dummy | Yes | Yes | Yes | Yes | Yes | Yes |
| Constant | 217.795 *** (21.158) | 218.924 *** (20.781) | 218.381 *** (20.828) | 218.839 *** (20.907) | 218.740 *** (21.029) | 136.783 *** (14.849) |
| Observations | 3987 | 3987 | 3987 | 3987 | 3987 | 3987 |
| Within-$R^2$ | 0.712 | 0.714 | 0.714 | 0.713 | 0.713 | 0.720 |

Notes: Standard errors in parentheses are clustered by prefecture-level city. *** $p < 0.01$, ** $p < 0.05$, * $p < 0.1$.

Column (4) considers an alternative definition of high-speed rail that includes ordinary railways also served by the C, D and G prefix bullet trains [16,19]. The estimates are again statistically significant for core cities and insignificant for non-core cities, and the magnitudes are quite similar to the baseline regression in column (3) of Table 2. Column (5) considers a more rigorous definition of high-speed rail that includes train services with peak speeds of over 300 km/h [1]. The estimates show that a more rigorous definition of HSR does not affect the robustness of our results. In column (6), all continuous variables

are winsorized at the 1st and 99th percentiles to minimize the effect of outliers. We again find similar magnitudes, suggesting that our results are not driven by outliers.

In sum, the results presented in Tables 2 and 3 indicate that HSR operations have gained a large market share and produced benefits for the dominant regions, consistent with the theoretical prediction of NEG models [5] and the empirical work of Baum-Snow et al. [6] on highways.

### 5.3. Heterogenous Treatment Effects among Non-Core Cities

High-speed rail cities are not equally distant to core cities. Some are close to core cities, while others are farther away. Therefore, Table 4 examines whether the effect of HSR varies by geographical proximity to core cities. Column (1) divides non-core HSR cities into core cities' geographically contiguous neighbors and other cities. We find a positive effect for farther away non-core HSR cities, with a magnitude that is about two times that of core cities' contiguity neighbors. Column (2) further divides the contiguity neighboring cities into neighbors of municipalities and neighbors of provincial capitals. The interaction coefficient for neighbors of municipalities is much larger and statistically significant at the 5% level, while the effect of HSR on GDP growth in provincial capitals' contiguity neighboring cities is close to zero. One possible explanation is that labor and investment are diverted from provincial capitals' neighbors to provincial capitals after a high-speed rail installation.

Column (3) investigates the possible heterogenous effects of high-speed rail by distance to core cities. We calculate the great-circle distance from the centroid of each high-speed rail city to its nearest core cities. The latitude and longitude information come from the National Geomatics Center of China. Given that the closest prefecture-level cities to municipalities exceed 50 km, we divide non-core high-speed rail cities into HSR cities within 100 km of core cities, HSR cities with distance to core cities from 100 km to 200 km, and HSR cities at least 200 km away from core cities. The estimates of HSR operation effects on GDP growth are negative for cities within 100 km of core cities and positive for cities 100 km away from core cities. Column (4) further divides non-core cities into various distances to municipalities and provincial capitals. We find that HSR operation decreases the annual GDP growth rate in cities within 100 km of provincial capitals by 0.35 percentage points, while the non-core cities farther away from provincial capitals are positively affected by HSR operation. Meanwhile, the estimated coefficients for HSR cities within 100 km of municipalities and HSR cities with a distance to municipalities from 100 km to 200 km are 0.79 and 0.68 respectively, suggesting that the HSR cities relatively close to municipalities benefited from municipalities' positive spillovers compared with HSR cities that are farther away, offseting the divert effect from core cities.

In sum, high-speed rail cities within 100 km of municipalities benefit more than those beyond 100 km, while non-core cities within 100 km of provincial capitals are negatively affected by HSR operation. One possible explanation is that the attractiveness of municipalities may decline when they become too large. Expensive land and housing and efficiency loss due to congestion may force some firms and workers to relocate to neighboring HSR cities. However, most provincial capitals may not develop enough to generate spillover effects [41]. Agglomeration shadows [12] are likely to play a role in attracting economic activity from nearby HSR cities to provincial capitals.

**Table 4.** Heterogenous treatment effects among non-core cities.

| | (1) | (2) | (3) | (4) |
|---|---|---|---|---|
| | **GDP Growth Rate** | **GDP Growth Rate** | **GDP Growth Rate** | **GDP Growth Rate** |
| Core city × HSRail | 1.829 *** (0.464) | | 1.806 *** (0.455) | |
| Municipality × HSRail | | 2.656 ** (1.182) | | 2.692 ** (1.176) |
| Provincial capital × HSRail | | 1.681 *** (0.456) | | 1.673 *** (0.450) |
| Wcore city × HSRail | 0.255 (0.329) | | | |
| Wmunicipality × HSRail | | 0.957 ** (0.436) | | |
| Wprovincial capital × HSRail | | 0.122 (0.343) | | |
| Wcore city (0–100 km) × HSRail | | | −0.217 (0.430) | |
| Wcore city (100–200 km) × HSRail | | | 0.704 ** (0.280) | |
| Wmunicipality (0–100 km) × HSRail | | | | 0.785 (0.801) |
| Wmunicipality (100–200 km) × HSRail | | | | 0.676 * (0.388) |
| Wprovincial capital (0–100 km) × HSRail | | | | −0.348 (0.455) |
| Wprovincial capital (100–200 km) × HSRail | | | | 0.595 ** (0.286) |
| Other cities × HSRail | 0.434 (0.289) | 0.426 (0.288) | 0.394 (0.343) | 0.379 (0.344) |
| log (GDP$_{t-1}$) | −15.553 *** (1.448) | −15.443 *** (1.450) | −15.469 *** (1.435) | −15.328 *** (1.440) |
| Open | 0.006 (0.204) | 0.003 (0.203) | 0.015 (0.198) | 0.015 (0.198) |
| Subway | 0.214 * (0.118) | 0.205 * (0.119) | 0.211 * (0.116) | 0.191 (0.118) |
| Share of industry in GDP | 0.309 *** (0.041) | 0.307 *** (0.041) | 0.308 *** (0.041) | 0.308 *** (0.041) |
| Share of services in GDP | 0.133 *** (0.042) | 0.131 *** (0.042) | 0.131 *** (0.042) | 0.129 *** (0.041) |
| Control × Year dummy | Yes | Yes | Yes | Yes |
| Constant | 218.121 *** (20.868) | 216.706 *** (20.901) | 217.054 *** (20.698) | 215.064 *** (20.779) |
| Observations | 3987 | 3987 | 3987 | 3987 |
| Within-$R^2$ | 0.713 | 0.713 | 0.714 | 0.714 |

Notes: Standard errors in parentheses are clustered by prefecture-level city. *** $p < 0.01$, ** $p < 0.05$, * $p < 0.1$.

### 5.4. The Effects of HSR on GDP Growth within Cities

The preceding section focuses on the HSR effects across cities, while this section examines the HSR effects within cities. In Table 5, we use data from prefectures with urban areas and examine whether high-speed rail increases the GDP growth rate within cities. Column (1) excludes the observations in 2017 so that the estimates can be compared with those from the total city. The results show that the HSR causes a 1.00 percentage point increase in GDP growth in the urban areas of core cities. HSR does not significantly affect economic growth in urban areas in non-core cities. Column (2) includes the observations in 2017. The regression results are consistent with column (1) in terms of significance and magnitudes. Column (3) replicates column (1) but includes the districts under cities with a change in land area of more than 1% each year. Once again, we find an estimate very close to the baseline in column (1). Column (4) replaces the binary HSR operation indicator by

the count of HSR lines. The estimation results suggest that the improving connectivity in the HSR network has caused a significant increase in GDP growth among urban areas in core cities. Columns (5) and (6) consider alternative definitions of high-speed rail, including railways with C, D and G prefix bullet trains and train services with peak speeds of over 300 km/h. We find no significant changes when using different measures of HSR.

To summarize, we find a robust positive effect of HSR on economic growth in core cities. However, the interaction coefficients for core cities are much smaller compared with the corresponding regression results based on urban and rural areas (total city), as shown in column (3) of Table 2 and columns (2), (4) and (5) of Table 3, implying that high-speed rail has triggered a relatively slower growth rate in the central cities and a process of suburbanization and decentralization within regional primate cities, consistent with the prediction of urban economics and the empirical results of Baum-Snow et al. [9] on highways and Navarro and Turner [11] on subways.

**Table 5.** The effects of HSR on GDP growth within cities.

| | (1) | (2) | (3) | (4) | (5) | (6) |
|---|---|---|---|---|---|---|
| | GDP Growth Rate | GDP Growth Rate | GDP Growth Rate | GDP Growth Rate | GDP Growth Rate | GDP Growth Rate |
| Core city × HSRail | 0.998 * | 1.145 ** | 1.003 * | | 1.069 * | 1.311 ** |
| | (0.558) | (0.559) | (0.528) | | (0.544) | (0.568) |
| Non-core city × HSRail | 0.158 | 0.200 | 0.061 | | 0.254 | 0.712 * |
| | (0.350) | (0.346) | (0.325) | | (0.334) | (0.402) |
| Core city × HSRail lines | | | | 1.379 ** | | |
| | | | | (0.605) | | |
| Non-core city × HSRail lines | | | | 0.531 | | |
| | | | | (0.438) | | |
| log (GDP$_{t-1}$) | −18.103 *** | −17.880 *** | −17.619 *** | −18.113 *** | −18.141 *** | −18.125 *** |
| | (1.724) | (1.709) | (1.592) | (1.716) | (1.714) | (1.723) |
| Open | 0.023 | 0.004 | 0.043 | | 0.120 | 0.347 |
| | (0.246) | (0.242) | (0.218) | | (0.232) | (0.258) |
| Subway | 0.057 | 0.026 | −0.003 | −0.035 | 0.062 | 0.013 |
| | (0.135) | (0.134) | (0.131) | (0.143) | (0.136) | (0.141) |
| Share of industry in GDP | 0.335 *** | 0.318 *** | 0.301 *** | 0.333 *** | 0.335 *** | 0.336 *** |
| | (0.057) | (0.055) | (0.052) | (0.056) | (0.056) | (0.057) |
| Share of services in GDP | 0.130 ** | 0.114 ** | 0.110 ** | 0.128 ** | 0.131 ** | 0.132 ** |
| | (0.059) | (0.057) | (0.052) | (0.058) | (0.058) | (0.059) |
| Control × Year dummy | Yes | Yes | Yes | Yes | Yes | Yes |
| Constant | 239.831 *** | 237.697 *** | 235.670 *** | 239.817 *** | 240.455 *** | 240.376 *** |
| | (23.656) | (23.364) | (21.696) | (23.572) | (23.506) | (23.660) |
| Observations | 3413 | 3641 | 4230 | 3413 | 3413 | 3413 |
| Within-$R^2$ | 0.620 | 0.626 | 0.625 | 0.621 | 0.620 | 0.621 |

Notes: Standard errors in parentheses are clustered by prefecture-level city. *** $p < 0.01$, ** $p < 0.05$, * $p < 0.1$.

## 6. Conclusions

In this paper, we examine the effects of HSR operation on the urban growth disparity and location of economic activity within and across cities. We find evidence of significant heterogeneity in the HSR operation effects on urban growth and TFP. The HSR effects are especially strong in municipalities and provincial capitals, whereas the effects are much weaker in non-core cities. Meanwhile, high-speed rail cities within 100 km of municipalities gain compared with those beyond 100 km, while non-core cities within 100 km of provincial capitals are negatively affected by HSR operation. We further corroborate that the introduction of HSR promotes the process of local decentralization within core cities. The policy implications include the following:

First, for the cities in the "agglomeration shadow" area, especially HSR cities close to provincial capitals, local governments should explore local resource advantages, avoid

competition with regional primate cities in tradable goods sectors that deviate from their comparative advantages. However, core cities and cities close to municipalities should seize the opportunities brought by high-speed rail, increase the supply of construction land and attract the inflow of factors that will promote the sustainable development of cities. Second, firms should pay attention to the change in the relative attraction of location and the flow of labor caused by high-speed rail. The suburban areas of core cities and the HSR cities close to municipalities should be noticed.

This paper differs from existing high-speed rail work in two main ways. First, we investigate the HSR effects within and across cities, which provide a broad picture of the spatial distributional effects of HSR. Second, the focus on growth effects provides a better understanding of the changes in share and economic geography. The conclusion of this paper is consistent with the theoretical prediction of the central peripheral model of New Economic Geography and the classic monocentric model of urban economics in that the HSR operation promotes centralization across cities and decentralization within regional primate cities.

Our study has several limitations that can inform future research. Due to the limitation of data availability, it is difficult for us to conduct more robustness checks and identify the mechanisms. For example, annual resident population data at the city level are not available in China. Another limitation is that the timing and placement of HSR are not assigned at random, making it more difficult to attribute urban growth to the introduction of HSR. Finally, distinguishing between growth and reorganization in terms of the effects of the transport improvement seems an obvious direction for further research.

**Author Contributions:** Conceptualization, H.Z.; methodology, H.Z.; validation, X.Y.; formal analysis, Y.C.; data curation, X.Y. and W.Z.; writing—original draft preparation, H.Z. All authors have read and agreed to the published version of the manuscript.

**Funding:** This research was funded by the National Social Science Fund of China, grant number 18CJL034.

**Institutional Review Board Statement:** Not applicable.

**Informed Consent Statement:** Not applicable.

**Data Availability Statement:** The data presented in this study are available on request from the authors.

**Conflicts of Interest:** The authors declare no conflict of interest.

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
