# Peer review of "High-Speed Rail and Urban Growth Disparity: Evidence from China"

_sustainability, doi:10.3390/su14138170_

Round 1

Reviewer 1 Report

  • The paper an appropriate length.
  • The key messages short, accurate and clear.
  • The text’s meaning is clear.
  • A well-written the introduction.
  • Sets out the argument ....
  • Summarizes recent research related to the topic ...
  • Highlights gaps in current understanding or conflicts in current knowledge ...
  • Establishes the originality of the research aims by demonstrating the need for investigations in the topic area.
  • Original and topicality can only be established in the light of recent authoritative research.
  • This research has enough data points to make sure the data are reliable.
  • The results seem plausible, the trends you can see support the paper's discussion and conclusions, There are sufficient data.
  • The references relevant, recent, and readily retrievable.
  • The author has a deep understanding of the paper's content.
  • The research has most interesting data.
  • Abstract highlight the important findings of the study.
  • The authors presenting findings that challenge current thinking.
  • The evidence they present strong enough to prove their case.
  • The correct references cited.

Reviewer 2 Report

In this paper it is examined the effects of high-speed rail (HSR) operation on urban growth disparity in China.

The paper presents several strengths, of which it is listed:

  1. The topic is interesting, and the paper is comprehensive and well written.
  2. The paper is well documented, and bibliographic references are comprehensive. The literature review is consistent and relevant.
  3. The contributions of the paper are consistent.
  4. The results seem coherent and described with sufficient clarity.
  5. Based on the results obtained generic and fundamental conclusions are drawn.
  6. There are premises for the development of research in this topic.

Consequently, the paper may be recommended for publication in present form.

Reviewer 3 Report

The study analyzes the relationship between high-speed rail and urban growth disparity in China examining 285 prefecture level cities with urban and rural areas (total city) over the period 2005-2019. Despite the high quality data used, the study could benefit from better defining research questions, stating the main goal more clearly and revising several methodological shortcomings. Detained suggestions include the following.

1.    Overall, the quality of writing should be improved. There are many unclear statements and grammatical errors in many places. In general, I recommend that the authors should work with a native-speaking editor to clarify the phrasing and word choices. This will make the manuscript much easier to absorb by people around the world who read English.
The part of introduction does not illustrate clearly the novelty of the specific study. Authors should clearly highlight the innovative target of their study. Please clearly list research gaps and contributions of this paper in introduction. This will focus readers on the research significance of this paper.
Literature part needs a deep restructure. While it highly covers the topic, it'd be better if short and focused paragraphs were used. It would also be better if the writing was organized around ideas rather than simply stating what each study did and found. Also, a summarized table with the most important previous empirical studies and findings would improve the quality of the paper. Author(s) should highlight how this study will be more useful the other researchers and scholars.
Regarding the empirical part, several issues should be addressed. Authors should clearly denote the econometric specifications they followed highlighting its advantages. There is a table that present baseline models without explaining what models are they. For instance, if fixed effects were selected, authors should mention the reason that this kind of analysis was employed. Did Hausman test take place.
Variables present several outliers such as the GDP growth that ranges from -19 to 37. Trimmed analysis should take place for the robustness of the results.
The inclusion of lag GDP should create endogeneity issues in the models. I am not sure if panel endogeneity was addressed.
In the Conclusion part, limitations and specific policy implications are missing. Especially policy implications should be connected with the empirical findings.

Reviewer 4 Report

The topic is very interesting and wide-ranging.

The section “Conclusions” should provide both scientific and practical implications. Then... are there any important limitations in your findings or methods used that point for the need of further research? (the section "conclusions" should mention some limitations of this work, as the author do not mention any limitation).

Finally, add the sources in Figures 1 and 2.

Round 2

Reviewer 3 Report

Authors responded in all comments adequately.